# Modeling Soil Nitrogen Content in South Patagonia across a Climate Gradient, Vegetation Type, and Grazing

**Pablo L. Peri [1,2,\*]**, **Yamina M. Rosas [3]**, **Brenton Ladd [4,\*]**, **Santiago Toledo [2]**,
**Romina G. Lasagno [1] and Guillermo Martínez Pastur [3]**

[1]  Instituto Nacional de Tecnología Agropecuaria (INTA), 9400 Río Gallegos, Argentina;
    lasagno.romina@inta.gob.ar
[2]  Universidad Nacional de la Patagonia Austral (UNPA)—CONICET, 9400 Río Gallegos, Argentina;
    toledo.santiago@inta.gob.ar
[3]  Laboratorio de Recursos Agroforestales, Centro Austral de Investigaciones Científicas (CADIC CONICET),
    9410 Ushuaia, Argentina; yamicarosas@gmail.com (Y.M.R.); cadicforestal@gmail.com (G.M.P.)
[4]  Escuela de Agroforestería, Universidad Científica del Sur, Lima 33, Perú
\*  Correspondence: peri.pablo@inta.gob.ar (P.L.P.); bladd@cientifica.edu.pe or brenton.ladd@gmail.com (B.L.)

**Abstract:** Soil total nitrogen (N) stock in rangelands, shrublands, and forests support key ecological functions such as the capacity of the land to sustain plant and animal productivity and ecosystem services. The objective of this study was to model soil total N stocks and soil C/N ratio from 0–30 cm depth across the region using freely accessible information on topography, climate, and vegetation with a view to establishing a baseline against which sustainable land management practices can be evaluated in Southern Patagonia. We used stepwise multiple regression to determine which independent variables best explained soil total N variation across the landscape in Southern Patagonia. We then used multiple regression models to upscale and produce maps of soil total N and C/N across the Santa Cruz province. Soil total N stock to 30 cm ranged from 0.13 to 2.21 kg N m$^{-2}$, and soil C/N ratios ranged from 4.5 to 26.8. The model for variation of soil total N stock explained 88% of the variance on the data and the most powerful predictor variables were: isothermality, elevation, and vegetation cover (normalized difference vegetation index (NDVI)). Soil total N and soil C/N ratios were allocated to three categories (low, medium, high) and these three levels were used to map the variation of soil total N and soil C/N ratios across Southern Patagonia. The results demonstrate that soil total N decreases as desertification increases, probably due to erosional processes, and that soil C/N is lower at low temperatures and increased with increasing precipitation. Soil total N and soil C/N ratios are critical variables that determine system capacity for productivity, especially the provisioning ecosystem services, and can serve as baselines against which efforts to adopt more sustainable land management practices in Patagonia can be assessed.

**Keywords:** soil total nitrogen; rangeland; livestock; climate; native forest; land-use

## 1. Introduction

Nitrogen (N) is one of the most limiting nutrients for plant growth in temperate and arid ecosystems. Nitrogen dynamics (mineralization or plant N uptake) influence important ecosystem functions (biogeochemical cycles and productivity), and structural components (species composition and biodiversity) of these ecosystems [1,2]. The amount of nitrogen stored in soil is related to climate through biotic processes associated with productivity of vegetation and decomposition of organic matter [3,4]. Other factors, particularly rainfall input, dry deposition input, nitrogen fixation,

and losses of inorganic nitrogen due to leaching contribute to the variability of nitrogen in soil [5–7]. Furthermore, nitrogen strongly influences the below-ground carbon cycling due to its effects on the decomposition of organic matter [8]. However divergent results have been reported, with positive [9], negative [8], or neutral [10] effects of N on soil organic carbon (SOC) decomposition. Lower C/N ratios in soils may increase net N mineralization by reducing microbial demand for N (microbial immobilization) during decomposition [11,12].

Southern Patagonia extends from latitudes 46° to 52°30′S and temperate grassland is the predominant ecosystem type, covering 85% of the total landscape. These grasslands contain a significant and variable amount of small shrub vegetation and could thus also be characterized as cold-arid steppe or rangeland. Native *Nothofagus* forest and woodland (mainly *Nothofagus pumilio* and *Nothofagus antarctica*) cover a narrow (100 km wide) and very long (1000 km) band of land on the mountainous land on the western border of the country and province. Disturbances that include livestock grazing and forestry heavily influence these arid ecosystems through the modification of vegetation cover, soil physical properties, microbial communities, carbon cycling, nitrogen fixation, and hydrologic processes [13,14]. Over the last 70 years, pronounced degradation of rangeland in Southern Patagonia has occurred due to high stocking rates, failure to use fallow or recovery periods, and because of poor management of livestock distribution across the landscape [15]. Grazing has increased bare ground and caused significant changes in plant species composition across the landscape [16,17], consistent with reviews that demonstrate that overgrazing may reduce aboveground net primary production (ANPP) and change floristic composition in rangelands [18,19]. In this context, rangeland livestock production, silvopastoral systems, or forest harvesting using sustainable management practices are essential to support increasing human populations and lifestyles. There is a definite need to develop quantitative measures of rangeland condition that can reflect the impacts of unsustainable grazing and poor land management practices in a way that is not subjective. In this context, soil quality and especially soil nitrogen (N) and the C/N ratio, have been proposed as integrative indicators of environmental quality, food security, and economic viability [20,21]. However, these indicators have not been assessed for Southern Patagonia.

Grazing intensity on extensively managed rangelands may affect ecosystem soil total N stocks and C/N ratios [22–25]. Previously, it has been documented in arid zones the importance of measuring grazing together with other environmental variables as drivers of soil C [26]. In Patagonia, the influence of grazing on soil N and soil C/N interacting with environmental factors remains poorly understood, despite the vast area and the economic importance of grazing for the region.

Here we (i) upscale measurements of soil total N stocks to 30 cm to the regional scale using multiple regression that uses topographic, climatic, and vegetation indices as independent variables, (ii) establish a regional baseline for soil N and soil C/N. We hypothesized that (1) soil total N would be lower and C/N higher where soil water content was high (due to leaching), (2) soil total N would be higher and C/N lower where temperatures were low (due to reduced decomposition), and (3) that the effects of physical environmental conditions (moisture, temperature, topography) would be stronger than the effects of grazing (stocking rates) on soil total N stocks in Southern Patagonia.

## 2. Material and Methods

The study used data from the PEBANPA (Parcelas de Ecología y Biodiversidad de Ambientes Naturales en Patagonia Austral—biodiversity and ecological long-term plots in Southern Patagonia) network of permanent plots [27]. For this study, 227 (Figure 1) environmental conditions and ecological characteristics of the PEBANPA plots are described in greater detail in Peri et al. [27].

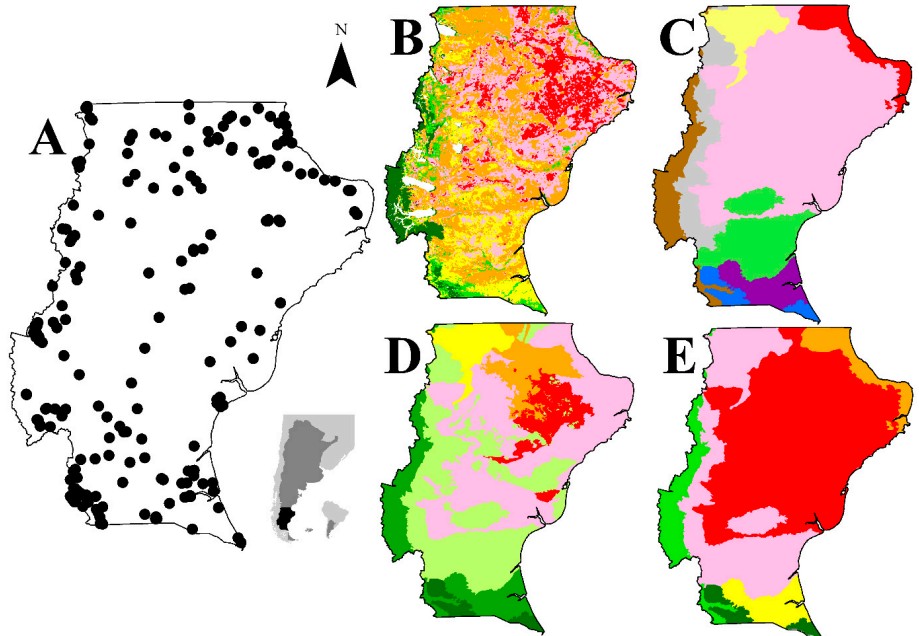

**Figure 1.** Characterization of the study area: (**A**) sample sites (black dots) in the Parcelas de Ecología y Biodiversidad de Ambientes Naturales en Patagonia Austral—biodiversity and ecological long-term plots in Southern Patagonia (PEBANPA) permanent plot network, location of Argentina (dark grey) and Santa Cruz province (black); (**B**) desertification (dark green = none, grey = slight degraded, yellow = moderate desertification, orange = moderate to severe desertification, pink = severe desertification, red = very severe desertification [28]; (**C**) main ecological areas (violet = dry Magellanic grass steppe, pink = central plateau; brown = Andean vegetation, yellow = mountains and plateaus, grey = sub-Andean grassland, blue = humid Magellanic grass steppe, green = mata negra thicket, red = shrub steppe San Jorge Gulf; (**D**) potential stocking rate (ewes/ha/year) (red ≤ 0.10, pink = 0.10–0.20, orange = 0.13–0.25, yellow = 0.16–0.30, light green = 0.20–0.30, green = 0.30–0.40, dark green ≥ 0.40); (E) actual stocking rate (ewes/ha/year) (red = 0.07, pink = 0.13, orange = 0.18, yellow = 0.22, green = 0.30, dark green = 0.50) (SIT–Santa Cruz).

*2.1. Soil Total Nitrogen Content*

For all 227 sites we extracted data of nitrogen concentration (%N) and soil bulk density (BD) from the PEBANPA database during 2003 and 2004 (see Peri et al. [27] for details of the methodology). Prior to analysis of N content, soil samples were finely milled to below 2 μm using a tungsten-carbide mill. Measurements of %N were made using a LECO auto-analyzer (St. Joseph, USA) at the Australian Nuclear Science and Technology Organization (ANSTO). Soil BD was estimated using the cylindrical core method (n = 3) by collecting a known volume of soil using a metal tube pressed into the soil (intact core), and determining the weight after drying. Knowing soil BD and using a soil depth of 0.3 m (Z), we calculated the soil nitrogen stock:

$$\text{Soil Nitrogen Stock (kg m}^{-2}) = \text{Nitrogen (g kg}^{-1}) \times \text{BD (Mg m}^{-3}) \times \text{Z (m)} \tag{1}$$

*2.2. Geographical Information System (GIS) Derived Independent Variables*

We assessed 30 potential explanatory variables (Table 1) for soil total N stock and soil C/N ratio. These variables were rasterized at 90 × 90 m resolution using the nearest resampling technique in ArcMap 10.0 software [29]. These rasterized variables covered climate, topography, vegetation, and land-use (see Table 1).

The majority of the variables considered were climatic variables (n = 21) [30] that included temperature, precipitation, and indexes of annual, monthly, and seasonal variation in these variables.

We also assessed potential evapo-transpiration and an aridity index [31]. The topography variables (n = 6) we assessed were: elevation, slope [32], and aspect [33]. Also, Euclidean distance to lakes and rivers was calculated using shape files obtained from the system of territorial information for Santa Cruz Province (Sistema de Información Territorial, SIT-Santa Cruz). Finally, vegetation metrics (n = 3), including the normalized difference vegetation index (NDVI) [34], net primary productivity (NPP) for the year 2015 [35], and a desertification index [28] where assessed.

**Table 1.** Explanatory variables used in soil total nitrogen content analysis.

| Category | Description | Code | Unit | Data Source |
|---|---|---|---|---|
| Climate | mean annual temperature | AMT | °C | WorldClim [1] |
| | mean diurnal range | MDR | °C | WorldClim [1] |
| | isothermality | ISO | % | WorldClim [1] |
| | temperature seasonality | TS | °C | WorldClim [1] |
| | max temperature of warmest month | MAXWM | °C | WorldClim [1] |
| | min temperature of coldest month | MINCM | °C | WorldClim [1] |
| | temperature annual range | TAR | °C | WorldClim [1] |
| | mean temperature of wettest quarter | MTWEQ | °C | WorldClim [1] |
| | mean temperature of driest quarter | MTDQ | °C | WorldClim [1] |
| | mean temperature of warmest quarter | MTWAQ | °C | WorldClim [1] |
| | mean temperature of coldest quarter | MTCQ | °C | WorldClim [1] |
| | mean annual precipitation | AP | mm·years$^{-1}$ | WorldClim [1] |
| | precipitation of wettest month | PWEM | mm·years$^{-1}$ | WorldClim [1] |
| | precipitation of driest month | PDM | mm·years$^{-1}$ | WorldClim [1] |
| | precipitation seasonality | PS | % | WorldClim [1] |
| | precipitation of wettest quarter | PWEQ | mm·years$^{-1}$ | WorldClim [1] |
| | precipitation of driest quarter | PDQ | mm·years$^{-1}$ | WorldClim [1] |
| | precipitation of warmest quarter | PWAQ | mm·years$^{-1}$ | WorldClim [1] |
| | precipitation of coldest quarter | PCQ | mm·years$^{-1}$ | WorldClim [1] |
| | global potential evapo-transpiration | EVTP | mm·years$^{-1}$ | CSI [2] |
| | global aridity index | GAI | | CSI [2] |
| Topography | elevation | ELE | m.a.s.l. | DEM [3] |
| | slope | SLO | % | DEM [3] |
| | aspect cosine | ASPC | cosine | DEM [3] |
| | aspect sine | ASPS | sine | DEM [3] |
| | distance to water bodies | DWD | km | SIT Santa Cruz [4] |
| | distance to rivers | DR | km | SIT Santa Cruz [4] |
| Vegetation and land-use | normalized difference vegetation index | NDVI | dimensionless | MODIS [5] |
| | net primary productivity | NPP | gr C/m$^2$/year | MODIS [6] |
| | desertification | DES | degree | CENPAT [7] |
| | vegetation types | VT | dimensionless | SIT Santa Cruz [4] |
| | stocking rate | SR | ewe/ha/year | SIT Santa Cruz [4] |
| | carrying capacity | RF | ewe/ha/year | SIT Santa Cruz [4] |

[1] Hijmans et al. [30], [2] Consortium for Spatial Information (CSI) [31], [3] Farr et al. [32], [4] SIT–Santa Cruz, [5] ORNL DAAC [34], [6] Zhao and Running [35], [7] Del Valle et al. [28].

The GIS methods used here are described in greater detail in Peri et al. [14,26,27].

*2.3. Modeling and Data Analyses*

An initial assessment of the independent variables listed in Table 1 for predicting soil total N stocks and soil C/N ratios was done using Pearson correlation coefficients. This initial screening was performed to identify independent variables free from collinearity.

We then subjected the most powerful independent variables free from collinearity to stepwise multiple regressions for prediction of soil total N stock and C/N ratio. For these regression analyses we only included an independent variable if the Pearson correlation coefficient was free from collinearity and if it value had a *p* value of <0.05. One hundred steps were used for final model selection.

The robustness of the regression models was assessed via the standard error (SE) of estimation (the r$^2$-adj), which is the average of the difference between predicted versus observed values, and the mean absolute error (AE) defined as the average difference between predicted versus the observed absolute values (Statgraphics Centurion software, Statpoint Technologies, USA).

We also tested the robustness of the regression models by analyzing the mean and absolute errors (differences between observed and modeled values of N content expressed in kg m$^{-2}$). Finally, we compared soil total N across gradients of land-use intensity and physical environmental variables: (i) vegetation types, (ii) stocking rates, (iii) soil cover (bare soil, shrubs, dwarf-shrubs, grasses, herbs, trees) (see further methodological detail see Peri et al. [14]).

We extrapolated the soil N regression model to obtain a map of soil total N stocks across Santa Cruz province (Argentina). The variables (isothermality, elevation, and NDVI) derived from the multiple linear regression model were integrated into a geographical information system (GIS) using ArcMap 10.0 software [29]. A mask was then applied using the NDVI as a reference to eliminate values less than 0.05 from the soil total N map to exclude zones where glaciers, water, rock, or bare soil are predominant [36]. For the map of soil total N values were assigned to three categories: low (0.17–0.37), medium (0.38–0.50), and high (0.50–1.87 kg N m$^{-2}$). The limits of each soil N class were defined so that each category contained an equal quantity of pixels for the whole province.

For each soil N class (low, medium, and high) we calculated the mean values and standard deviation across 25 continuous variables (including climate, topographic, and landscape variables, see Table 1) using data from the entire province. In addition, mean values and standard deviation of soil N were calculated for the following discrete variables: animal stocking rate, carrying capacity for livestock (forage availability), vegetation type, and desertification.

Finally, the map of soil C/N ratio for the province of Santa Cruz was obtained by combining a SOC map previously reported [26] and the present soil N map using ArcMap 10.0 software [29]. For C/N ratios we used the following categories: low (4–9), medium (10–12), and high soil C/N ratio (13–27). Similarly, to the soil N analysis, mean values and standard deviation of 25 continuous variables (see Table 1) for each soil C/N class (low, medium, and high) were calculated. Also, mean values and standard deviation of soil C/N were calculated for the following discrete variables: animal stocking rate, carrying capacity for livestock, vegetation type, and desertification.

## 3. Results

Soil nitrogen concentration (%N) and soil total N stock to 30 cm depth ranged from 0.03% to 0.81% and from 0.13 to 2.21 kg N m$^{-2}$, respectively (Table 2). C/N values measured in the field varied between 4.5 and 26.8 (Table 2). Many independent variables correlated strongly with soil total N stock using the Pearson's correlation index (Appendix A), where NDVI (0.808, $p < 0.001$) was the most closely correlated. The variable distance to rivers (DR) was not significantly correlated with soil total N stock using the Pearson's correlation index (Appendix A).

**Table 2.** Soil nitrogen concentration (%N), soil total N stock, and C/N ratio (30 cm depth) measured in different ecosystems in Santa Cruz province, South Patagonia, Argentina.

| Vegetation Types | Soil N Concentration (%N) | Soil Total N (kg N m$^{-2}$) | Soil C/N |
|---|---|---|---|
| *Nothofagus* forest | 0.11–0.81 | 0.59–2.21 | 4.5–10.3 |
| Grasslands | 0.04–0.51 | 0.16–1.66 | 5.5–26.8 |
| Shrublands | 0.03–0.22 | 0.13–0.76 | 7.6–11.0 |
| Wetlands | 0.06–0.73 | 0.34–1.78 | 8.9–12.1 |

The stepwise multiple regression procedure identified a regression model with three independent variables: isothermality (ratio of average day variation in temperature divided by annual variability in temperature) (ISO, %), elevation (ELE, meters above sea level (m.a.s.l.)) and normalized vegetation index (NVDI, dimensionless).

When univariate correlations were performed in all three variables these variables correlated strongly with soil total N stock and there was no evidence of collinearity between them ($p < 0.001$).

The fitted model ($r^2$-adj = 0.884; F = 575.2; SE = 0.32; AE = 0.23) explained 88.4% of variation in soil total N stocks and had the following form:

$$N \text{ (kg m}^{-2}) = 1.60 \times NDVI + 0.0002 \times ELE + 0.001 \times ISO \tag{2}$$

This model had an average error of −0.01 kg N m$^{-2}$, and an absolute error of 0.24 kg N m$^{-2}$. When the performance of the model was tested across different vegetation types and management related gradients (e.g., stocking rates), it became apparent that the error dispersion wasn't homogeneous. In general, error increased with soil total N stock.

The map of the soil total N stocks across Santa Cruz exhibits a continuous decline from the mountainous zone to eastern limits of the Province where rangeland predominates (Figure 2). Also, soil total N values are high in river valleys and wetlands.

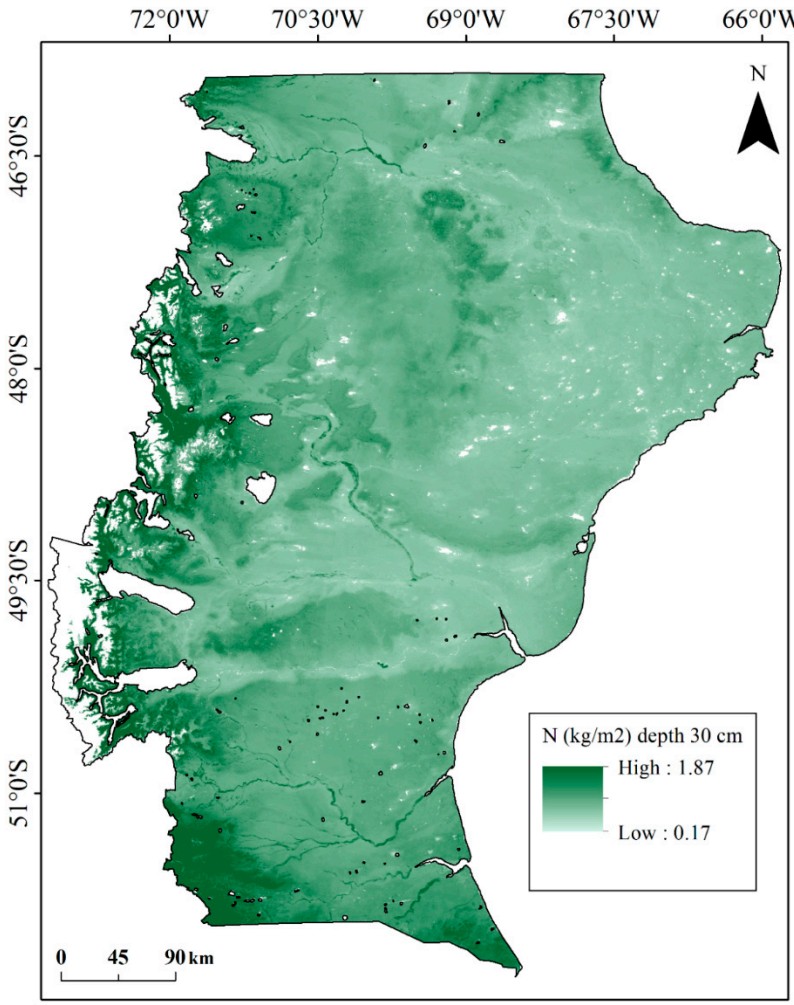

**Figure 2.** Soil total nitrogen content stock (30 cm depth) in Santa Cruz province, South Patagonia, Argentina.

Soil C/N increased from west to east, the opposite pattern to that observed for soil N stocks (Figure 3).

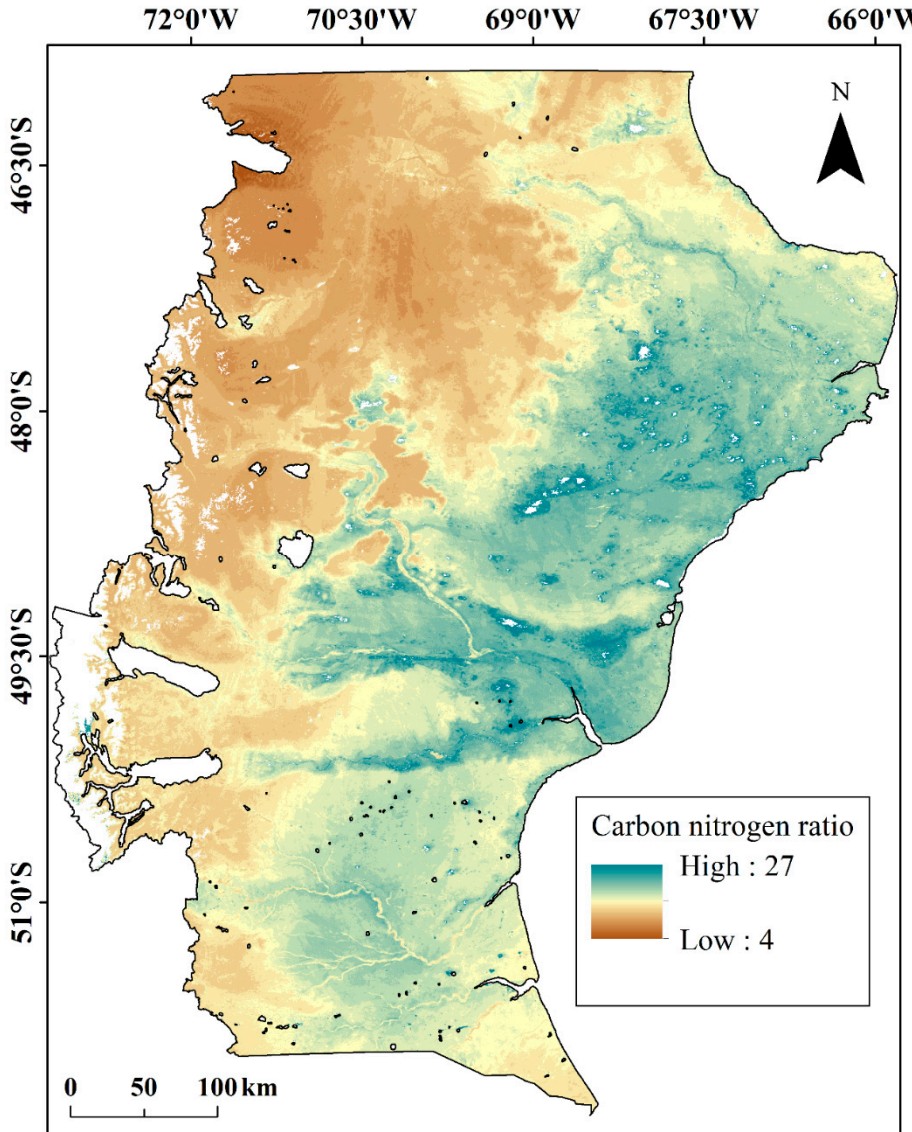

**Figure 3.** Soil C/N ratio in Santa Cruz province, South Patagonia, Argentina.

　　The most parsimonious regression models, identified by the stepwise regression procedures, contained only variables that characterized the physical environment (i.e., NDVI, isothermality, and elevation) (Table 3). In contrast, the variables related to human land-use intensity (i.e., stocking rates) had non-significant effects on soil total N stocks and soil C/N ratios across the Southern Patagonian landscape (Table 4).

**Table 3.** Mean (standard deviation) values of climatic, topographic, and vegetation variables classified according to the soil N classes. Total area represents values for the entire province, N content categories: low (0.17–0.37 kg N m$^{-2}$), medium (0.38–0.50 kg N m$^{-2}$), and high (0.51–1.87 kg N m$^{-2}$) values and C/N categories: low (4–9), medium (10–12), and high (13–27).

| Variable | | Total Area | N Content (kg N m$^{-2}$) | | | C/N | | |
|---|---|---|---|---|---|---|---|---|
| | | | Low | Medium | High | Low | Medium | High |
| Temperature | MAT | 7.8 (2.4) | 9.7 (1.1) | 8.1 (1.6) | 6.0 (1.9) | 6.9 (2.5) | 7.9 (2.1) | 8.9 (1.4) |
| | MDR | 10.3 (0.6) | 10.5 (0.4) | 10.6 (0.5) | 10.0 (0.7) | 10.3 (0.7) | 10.3 (0.7) | 10.5 (0.4) |
| | ISO | 46.4 (0.2) | 45.7 (1.06) | 45.8 (1.4) | 46.6 (1.7) | 45.3 (1.0) | 46.3 (0.2) | 46.4 (0.1) |
| | TS | 4.5 (0.4) | 4.7 (0.2) | 4.6 (0.3) | 4.2 (0.5) | 4.6 (0.4) | 4.4 (0.5) | 4.5 (0.3) |
| | MAXWM | 19.6 (3.2) | 21.9 (1.5) | 20.2 (2.2) | 17.3 (2.5) | 18.8 (3.3) | 19.6 (2.9) | 20.8 (1.8) |
| | MINCM | −2.7 (2.2) | −0.9 (1.4) | −2.7 (1.6) | −4.1 (1.8) | −3.7 (2.1) | −2.4 (2.0) | −1.6 (1.6) |
| | TAR | 22.2 (1.8) | 22.8 (1.1) | 22.9 (1.4) | 21.3 (1.8) | 22.6 (1.8) | 22.0 (1.9) | 22.4 (1.1) |
| | MTWEQ | 5.7 (2.9) | 5.0 (2.1) | 6.6 (3.1) | 5.8 (3.1) | 5.8 (3.3) | 6.4 (2.8) | 5.2 (2.4) |
| | MTDQ | 9.8 (3.7) | 11.2 (3.0) | 10.4 (3.8) | 8.4 (3.2) | 11.3 (3.1) | 9.6 (4.2) | 9.2 (2.9) |
| | MTWAQ | 13.2 (2.8) | 15.3 (1.3) | 13.7 (2.0) | 11.1 (2.2) | 12.6 (3.0) | 13.2 (2.6) | 14.3 (1.7) |
| | MTCQ | −2.65 (2.2) | 3.6 (1.1) | 2.0 (1.4) | 0.4 (1.7) | 0.9 (2.1) | 2.0 (1.8) | 2.9 (1.4) |
| Precipitation | AP | 245.9 (181.4) | 200.7 (41.0) | 193.6 (74.5) | 278.6 (166.5) | 258.7 (1.5) | 224.1 (120.7) | 195.5 (49.6) |
| | PWEM | 30.2 (18.9) | 25.3 (5.0) | 24.8 (8.6) | 33.7 (17.5) | 33.8 (15.2) | 27.0 (12.6) | 23.6 (5.4) |
| | PDM | 13.6 (12.5) | 11.1 (2.8) | 10.2 (5.1) | 15.1 (11.5) | 13.1 (10.3) | 12.4 (8.3) | 11.2 (3.4) |
| | PS | 24.4 (6.6) | 24.4 (4.4) | 25.2 (6.8) | 24.4 (7.7) | 29.8 (6.9) | 23.3 (5.4) | 21.4 (3.4) |
| | PWEQ | 79.8 (53.2) | 67.3 (14.8) | 63.4 (24.1) | 90.3 (50.8) | 87.1 (45.0) | 72.7 (36.5) | 63.1 (17.2) |
| | PDQ | 46.4 (41.1) | 37.4 (8.7) | 35.1 (16.2) | 51.7 (36.4) | 45.8 (32.4) | 42.0 (26.5) | 37.4 (10.6) |
| | PWAQ | 53.6 (42.9) | 44.4 (10.7) | 41.5 (18.1) | 59.9 (38.6) | 47.7 (34.7) | 50.9 (29.5) | 47.8 (11.8) |
| | PCQ | 67.3 (46.0) | 58.3 (13.9) | 53.2 (21.3) | 74.7 (45.3) | 72.8 (40.0) | 61.1 (31.9) | 53.9 (16.0) |
| | EVTP | 807.9 (101.6) | 880.5 (49.1) | 830.6 (73.5) | 733.6 (82.4) | 790.1 (106.41) | 803.2 (98.6) | 846.7 (60.4) |
| | GAI | 0.3 (0.4) | 0.2 (0.1) | 0.2 (0.1) | 0.4 (0.3) | 0.3 (0.3) | 0.3 (0.2) | 0.2 (0.1) |
| Topographic | ELE | 468.8 (383.8) | 204.0 (125.9) | 462.0 (244.1) | 664.3 (405.5) | 805.2 (298.4) | 366.9 (226.0) | 187.0 (106.0) |
| | SLO | 5.0 (5.8) | 3.5 (2.9) | 4.2 (4.2) | 6.3 (6.5) | 6.4 (6.3) | 4.3 (4.6) | 3.3 (2.9) |
| Landscape | NDVI | 0.2 (0.1) | 0.1 (0.0) | 0.2 (0.0) | 0.3 (0.1) | 0.3 (0.2) | 0.2 (0.1) | 0.2 (0.0) |
| | NPP | 128.0 (68.5) | 104.7 (39.0) | 110.3 (40.9) | 157.5 (99.5) | 118.9 (86.9) | 143.1 (76.3) | 113.0 (39.0) |

MAT = mean annual temperature (°C), MDR = mean diurnal range (°C), ISO = isothermality (%), TS = temperature seasonality (°C), MAXWM = maximum temperature of warmest month (°C), MINCM = minimum temperature of coldest month (°C), TAR = temperature annual range (°C), MTWEQ = mean temperature of wettest quarter (°C), MTDQ = mean temperature of driest quarter (°C), MTWAQ = mean temperature of warmest quarter (°C), MTCQ = mean temperature of coldest quarter (°C), AP = mean annual precipitation (mm·year$^{-1}$), PWEM = precipitation of wettest month (mm·year$^{-1}$), PDM = precipitation of driest month (mm·year$^{-1}$), PS = precipitation seasonality (%), PWEQ = precipitation of wettest quarter (mm·year$^{-1}$), PDQ = precipitation of driest quarter (mm·year$^{-1}$), PWAQ = precipitation of warmest quarter (mm·year$^{-1}$), PCQ = precipitation of coldest quarter (mm·year$^{-1}$), EVTP = global potential evapo-transpiration (mm·year$^{-1}$), GAI = global aridity index, ELE = elevation (meters above sea level (m.a.s.l.)), SLO = slope (°). NDVI = normalized difference vegetation (dimensionless), NPP = net primary productivity (gr C/m$^2$/year).

**Table 4.** Mean values (standard deviation, SD) and areas of N (kg N m$^{-2}$) at 30 cm depth and C/N content as a function of the discrete independent variables.

| Variable | Category | Total Area (km$^2$) | Mean (SD) | |
| --- | --- | --- | --- | --- |
| | | | N | C/N |
| Desertification | No desertification | 9737.2 | 0.89 (0.35) | 9.39 (1.83) |
| | Slight degraded | 11866.5 | 0.79 (0.27) | 9.43 (2.06) |
| | Moderate desertification | 33816 | 0.60 (0.20) | 10.89 (2.23) |
| | Moderate to severe desertification | 83417.1 | 0.48 (0.15) | 10.69 (2.88) |
| | Severe desertification | 62929.6 | 0.40 (0.12) | 11.81 (2.89) |
| | Very severe desertification | 29178.5 | 0.32 (0.08) | 13.19 (2.85) |
| Vegetation types | *Nothofagus* Forest | 10107.3 | 1.03 (0.34) | 9.09 (1.34) |
| | Humid Magellanic grass steppe | 6049.9 | 0.88 (0.16) | 10.57 (1.08) |
| | Sub-Andean grassland | 18797.8 | 0.71 (0.18) | 8.05 (1.59) |
| | Dry Magellanic grass steppe | 11786.8 | 0.58 (0.09) | 12.25 (1.11) |
| | Mata negra thicket | 28309.4 | 0.50 (0.10) | 12.39 (1.24) |
| | Mountains and plateaus | 13089.2 | 0.49 (0.13) | 7.26 (1.35) |
| | Shrub steppe San Jorge Gulf | 11905.1 | 0.41 (0.10) | 10.95 (1.72) |
| | Central plateau | 130855.4 | 0.39 (0.11) | 11.95 (3.01) |
| Potential stocking rate (ewes/ha/year) | <0.10 | 16908.3 | 0.33 (0.07) | 13.84 (2.33) |
| | 0.10–0.20 | 83587.5 | 0.39 (0.11) | 12.12 (3.01) |
| | 0.13–0.25 | 18617.1 | 0.41 (0.11) | 9.79 (1.80) |
| | 0.16–0.30 | 13140.7 | 0.49 (0.12) | 7.27 (1.34) |
| | 0.20–0.30 | 71046.8 | 0.54 (0.17) | 10.83 (2.57) |
| | 0.30–0.40 | 21793.7 | 0.79 (0.33) | 10.78 (1.99) |
| | >0.40 | 5916.1 | 0.89 (0.16) | 10.54 (1.09) |
| Actual stocking rate (ewes/ha/year) | 0.07 | 130735.6 | 0.39 (0.11) | 11.95 (3.01) |
| | 0.13 | 60086.7 | 0.56 (0.17) | 9.92 (2.72) |
| | 0.18 | 11801.8 | 0.41 (0.10) | 10.94 (1.71) |
| | 0.22 | 11705.9 | 0.58 (0.09) | 12.25 (1.10) |
| | 0.30 | 10094.6 | 1.04 (0.34) | 9.08 (1.33) |
| | 0.50 | 5886.3 | 0.89 (0.16) | 10.58 (1.09) |

There were differences in soil total N stocks across vegetation types with mean values that varied from 0.39 kg N m$^{-2}$ in grasslands on the central plateau to 1.03 kg N m$^{-2}$ in *Nothofagus* forest (Table 4). Soil C/N ratio ranged from 7.26 (mountains and plateaus) to 12.39 (shrublands dominated by mata negra). At high animal stocking rates, soil total N stocks were also relatively high, but soil C/N did not show a clear pattern of variation with stocking rate. Finally, soil total N stocks decreased with desertification (from 0.89 kg N m$^{-2}$ at sites with little desertification to 0.32 kg N m$^{-2}$ at sites where desertification was pronounced), probably reflecting erosion (Table 4). Soil C/N showed the opposite pattern and increased with desertification.

## 4. Discussion

Our model for soil total N stocks was able to account for 88% of the variation of this soil property across the study area with values ranging from 0.13 to 2.21 kg N m$^{-2}$. In the present study, soil total N stock to 30 cm and C/N ratio (the latter ranging from 4.5 to 26.8) was mainly a function of three variables: NDVI, isothermality, and elevation. All three variables can be obtained from freely available geographical datasets and this highlights the value of the rapidly evolving possibilities that GIS offers for establishing quantitative baselines against which we can monitor different strategies to improve sustainability of land management. Although the effect of stocking rate on soil N was minimal in this analysis it is conceivable that grazing may have indirectly affected soil N by modifying the type of vegetation cover [37]. In this study, vegetation cover, as represented by a normalized difference vegetation index (NDVI), was a strong predictor of soil total N stock. This was consistent with Kunkel et al. [38] who reported that NDVI predicted soil N distribution in semi-arid montane ecosystem. Further analysis, such as path analysis, may shed light on the interactions and underlying factors that resulted in the correlations observable in this analysis.

Soil N and soil C/N ratios are important variables that relate to key ecosystem services, especially provisioning services such as production of forage, meat, and fiber in this region. The results

of this analysis however indicate that for soil N and soil C/N ratios to be used as indicators of sustainable land management we need to use climatic and topographic variables as covariates, otherwise the influence of these potential covariates will likely swamp any effect of anthropogenic interventions.

The correlation between soil total N and elevation may be due to lower temperatures at higher elevations. Some models and studies suggest that high latitude (cold) forests and grasslands may behave as a N source in response to increased decomposition of soil organic matter or net nitrogen mineralization when temperatures increase [39–41]. In this sense, different studies have reported the direct effect of temperature on N mineralization with the highest rates at temperatures near 25 °C [42,43]. Bahamonde et al. [44] measured values of mineralization and nitrification in native forests of Southern Patagonia and demonstrated that low soil temperature is a limiting factor for N mineralization (<13 °C during October and February). Thus, the temperature sensitivity of decomposing organic matter in soil partly determines how much N will be transferred to the atmosphere because of global warming. The interaction of climate change with C pools in extremely temperate ecosystems may be particularly important because climate change is expected to be greatest for these ecosystems. In Southern Patagonia, mean maximum annual temperature is predicted to increase by 2–3 °C by 2080 between 46° and 52°30′ south latitude [45] and this will have significant effects on Patagonian ecosystems.

The effects of elevation on soil total N stocks may also have been driven by variation in rainfall. We suggest this possibility because the high elevation sites in Southern Patagonia receive more rainfall. Soil N may increase as rainfall increases. It has also been demonstrated that increased variability in rainfall and soil water content significantly affected soil N in grasslands [46,47]. Post et al. [1] reported that soil N storage ranged from 0.2 kg m$^{-3}$ in warm deserts to 2 kg m$^{-3}$ in rainy tundra, with a peak of 1.6 kg m$^{-3}$ in subtropical wet forests. Also, it has been demonstrated that N availability is strongly coupled with the quantity and quality of soil organic carbon pools and soil water availability [2]. Because soil N and water availability may covary across a precipitation gradient, it is difficult to separate the effects of plant production and water availability on variation in soil total N stocks. Thus, the strong and direct relationship between rainfall and soil total N may be related to ANPP and/or mean soil water content. In the present work net primary productivity (NPP) influenced positively soil N compared to the average for the province. This is consistent with Austin and Sala [47] who found that inorganic soil N was strongly correlated to both mean annual precipitation (MAP) and ANPP suggesting a direct control of precipitation on N turnover. Thus, it is clear that precipitation may constrain plant production in arid ecosystems. The results of studies that have assessed the correlation between soil moisture and N mineralization are variable. While Mazzarino et al. [48] and Xueling et al. [49] have reported positive correlations, other studies found no correlation [50,51]. These differences may relate to the effect of soil moisture on N mineralization and could be due to strong interactions between temperature and soil moisture affecting the process [52].

In this analysis soil total N stocks were relatively high above 462 m.a.s.l. (in mountain environments). This is consistent with Tashi et al. [53] along a transect from 317 to 3300 m.a.s.l. in the eastern Himalayas and with Shedayi et al. [54] in the Karakorum mountain range of Pakistan who reported that soil N stocks increased with altitude. This may be due to changes in climatic variables (mainly precipitation and temperature) and vegetation types along altitudinal gradients that influence in consequence the quality, quantity, and turnover of soil organic matter. Tashi et al. [53], using a meta-analysis, determined that the increase of soil N content with altitude was mainly influenced by increased N input to soils from forest canopies and the effect of temperature.

Vegetation types in Patagonia also influenced soil total N, being higher in forest than in grasslands or shrublands. Similarly, Tian et al. [55] reported that in the north of China, the highest N values occurred in the forests and woodlands (1668.2 g/m$^3$), followed by wetland soils (1459.9 g/m$^3$), and that dry land soil ranked the lowest (739.5 g/m$^3$). There are other examples of changes in soil N content according to vegetation type; for example, in the shrubby savanna of Cerrado, Brazil [56] and in Mediterranean natural areas [57]. This highlights the importance of a diverse range of environmental

conditions, mainly soil water and temperature, but also input of organic residues, soil microbial biomass, and soil properties on the magnitude of mineralization among different ecosystems.

Grazing also affected soil total N stocks in this analysis, but the effects were small and in the wider literature the effect of livestock grazing on N in rangeland soils is not well defined. Berg et al. [58] sampled eight pastures moderately grazed by cattle and eight adjacent enclosures ungrazed by livestock for 50 years on sandy rangeland in western Oklahoma, and found no differences in soil N concentration to a constant depth of 5 cm. N stocks declined in yak grazing by 27.4% compared with grazing-excluded sites (five years of grazing exclusion) at the eastern edge of the Qinghai–Tibetan Plateau [24]. Conant et al. [59] reported that effective grazing management techniques can increase above-ground biomass and have the potential to increase C and N storage in soil. In this regional study, sites where animal stocking rates are high, the soil total N also showed high values. This is logical and shows that grasslands with better soils have higher densities of livestock. Soils with low N content may represent areas where historical high stocking rates have decreased vegetation cover (and degraded soil and/or areas that have inherently low ANPP due to physical environmental constraints, such as suboptimal temperatures). This highlights the importance of establishing baselines that not only consider anthropogenic impacts but also environmental conditions. Quantitative indicators like soil total N stocks will vary systematically with variables like elevation, and any assessment of variation at the regional scale will require that these influences are constrained.

N is essential for plant growth, and the capacity of soils to supply N to plants is linked to the amount and nature of the soil organic matter. Peri et al. [14] reported that litter cover, litter depth, and soil carbon in the uppermost soil layers in the Magellanic grasslands, were lower under heavy long-term stocking rates than at sites under moderate grazing intensity. Increased nitrogen availability increases productivity and biomass accumulation substantially, at least over the short-term. Grassland ecosystems with high soil organic matter may promote organic matter decomposition (microbial activity) by continuous addition of litter and root turnover, thereby increasing soil respiration rates [3].

The soil C/N map developed for the entire study area demonstrated a mostly inverse pattern to that of soil N across the environmental and land-use variables. For example, soil C/N was lower at low temperatures, decreased with precipitation, and C/N did not show a clear pattern of variation with altitude, vegetation type, or grazing. In contrast, Piñeiro et al. [22] showed that C/N ratios frequently increased as grazing intensity increased, which suggests potential N limitations for soil organic matter formation under grazing. General trends suggested that increasing grazing pressure appeared to decrease soil organic carbon and increase the C/N ratio [23].

High C/N ratios could limit microbial activity due to lack of nitrogen. Gallardo and Schlesinger [60] reported a positive correlation between microbial biomass and soil C/N ratio to values less than 20. In this study, the highest soil C/N ratio of 12.39 was determined in the mata negra thicket in a very cold life zone. This may imply very low nitrification and mineralization rates. Rowe et al. [61] reported that acid grassland and heathland sites for which both C/N and nitrate flux measurements were available; deciduous woodland and acid grasslands typically had lower C/N ratios and began leaching nitrate at a lower C/N ratio than coniferous woodland and heathland. This may indicate that in the Patagonian region, low litter quality may result in high C/N ratios. According to the stoichiometric decomposition theory, the microbial activity is highest, and decomposition rates are maximal, if C and N input with substrate matches microbial demands, that is, when this input corresponds to a stoichiometric ratio of C to N above 20 [62].

## 5. Conclusions

Soil total nitrogen stocks across the sites analyzed here were influenced by a large number of variables: isothermality, elevation, and NDVI. Good management practices, such as low impact grazing or native sustainable harvesting in native forest can maintain or even increase soil total N stocks. However, economic incentives and policies are needed to maintain or increase soil N stocks. Understanding the causes of variation and mapping of soil total N and C/N ratio in Patagonia is

a first step which allows for assessment of the sustainability of land management at the regional scale. This may assist in increasing the resilience of rangeland and forest under climate change. This study contributes to solve regional ecological and socio-economic challenges in the sustainable use of our native ecosystems. However, research institutions and farmers must have a sustained commitment to finance and maintain these large unique long-term plots and research platforms. Also, there is a need to experiment with alternative forms of sustainable land management and mechanistic study to better understand the underlying factors that generate variation in soil total N stocks across this landscape.

**Author Contributions:** P.L.P. and G.M.P. conceived and designed the experiments, and wrote the paper; R.G.L. and S.T. performed the experiments; Y.M.R. and B.L. mainly analyzed the data and contributed analysis tools.

**Acknowledgments:** The present research was supported by the INTA and UNPA.

**Conflicts of Interest:** The authors declare no conflict of interest.

## Appendix A

**Table A1.** Pearson's correlation index among exploratory variables used in N carbon content (kg N m$^{-2}$) analysis. (See Table 1 for variable definitions).

| Category | Variables | N | |
|---|---|---|---|
| | | Correlation | *p*-value |
| Climate | AMT | −0.438 | <0.001 |
| | MDR | −0.3462 | <0.001 |
| | ISO | 0.1571 | =0.0164 |
| | TS | −0.4607 | <0.001 |
| | MAXWM | −0.4671 | <0.001 |
| | MINCM | −0.2628 | <0.001 |
| | TAR | −0.3872 | <0.001 |
| | MTWEQ | −0.2042 | =0.0017 |
| | MTDQ | −0.2796 | <0.001 |
| | MTWAQ | −0.4676 | <0.001 |
| | MTCQ | −0.3371 | <0.001 |
| | AP | 0.6334 | <0.001 |
| | PWEM | 0.6149 | <0.001 |
| | PDM | 0.6456 | <0.001 |
| | PS | −0.2415 | <0.001 |
| | PWEQ | 0.6192 | <0.001 |
| | PDQ | 0.6304 | <0.001 |
| | PWAQ | 0.627 | <0.001 |
| | PCQ | 0.6044 | <0.001 |
| | EVTP | −0.4839 | <0.001 |
| | GAI | 0.6632 | <0.001 |
| Topography | ELE | 0.3489 | <0.001 |
| | SLO | 0.4821 | <0.001 |
| | ASPC | −0.1291 | =0.049 |
| | ASPS | 0.232 | <0.001 |
| | DWB | −0.1855 | =0.0045 |
| | DR | −0.0953 | =0.1468 |
| Landscape and land-use | NDVI | 0.8082 | <0.001 |
| | PPN | 0.6865 | <0.001 |
| | DES | −0.4656 | <0.001 |

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
