# Peer review of "Modeling Soil Nitrogen Content in South Patagonia across a Climate Gradient, Vegetation Type, and Grazing"

_sustainability, doi:10.3390/su11092707_

Round 1

Reviewer 1 Report

Review of the manuscript: Modelling soil nitrogen content in South Patagonia across a climate gradient, vegetation type and grazing.

In think, this manuscript have to be accepted. In my opinion, this is a novel and necessary study in an interesting semi-arid area, which can be maybe extrapolate to other similar areas. In addition, the generation of maps with difference in the gradient of total N stocks and C/N gradient are a good illustration for readers and for easily recommend future and adequate sustainable management practices.

In general, the statistical analysis are enough correct, the introduction and discussion are good explained with clear language and supported with good references. The results are also good and structured in tables and figures.

However, I would like to recommend to the authors minimum modifications:

-          Please, in all the document add “total” when you talk about N: “Soil total nitrogen stock” because you have to be sure that people not confuse with other pools or forms of nitrogen. It is the general terminology used in this topic.

-          Line 87-89. Please re-write the hypothesis number 1 with short sentences, it is confuse.

-          In Material and methods is necessary that you write from with years are this dataset and if all the variables were measured for the same year.

-          Figure 1. Please use colors not with this grey-black-white.

-          Please add a table with the initial values of soil total nitrogen concentration, soil total nitrogen stock and C/N ratios in rangelands, shrubland and forest. Do you also have data of C? Why the authors don´t include also the organic C stocks?

Author Response

General comments

The comments by both referees have been noted and largely acted upon. The manuscript has been carefully revised. It has been improved by providing clarity, data analysis and interpretation. Also, sentences in section Discussion has been revised to avoid repetition with other publications.

Reviewers' comments:
Reviewer #1

-          In all the document “total” when mentioned N has been added. Now we wrote “Soil total nitrogen stock”.

-          Line 87-89. The hypothesis number 1 was re-written with short sentences and split in two hypotheses.

-          The years of dataset measurements have been specified.

-         Figure 1 now is with colors.

-          A table with the initial field values of soil total nitrogen concentration, soil total nitrogen stock and C/N ratios in rangelands, shrubland, wetlands and forest has been added. Data of C has been previously published in Peri et al. 2018.

Reviewer 2 Report

The authors have created an adequate model for N and C/N in Patagonia. Unfortunately, there are some annoying errors in the text. 1. Evaluation of content N probably overstated by an order of magnitude. 2. Values of C/N = 1 and 2 too low for this index toward soils. It should be recalculated or explained. 

But this mistaken in general are not influes on scientific values. Because regression analysis does not depend on absolute values in this case.

Remarks in the text.

Author Response

General comments

The comments by both referees have been noted and largely acted upon. The manuscript has been carefully revised. It has been improved by providing clarity, data analysis and interpretation. Also, sentences in section Discussion has been revised to avoid repetition with other publications.

Reviewer #2

Errors in the text and the magnitude of C/N values (low values) have been corrected.

Remarks in the text have been corrected.